# At the Intersection of Critical Care and Infectious Diseases: The Year in Review

**DOI:** 10.3390/biomedicines12030562

**Published:** 2024-03-02

**Authors:** Sarah R. Sabo, Aarthi Venkatramanan, Andrew F. Shorr

**Affiliations:** 1Department of Medicine, Medstar Washington Hospital Center, Washington, DC 20010, USA; sarahrosesabo@gmail.com (S.R.S.); aarthivenkatramanan@gmail.com (A.V.); 2Pulmonary and Critical Care Medicine, Medstar Washington Hospital Center, Washington, DC 20010, USA

**Keywords:** antibiotic, infection, intensive care unit, mechanical ventilation, pneumonia, prevention, resistance, quality, review, sepsis

## Abstract

Severe infection represents a leading reason for admission to the intensive care unit (ICU) while nosocomial infection can arise as a complication of care in the ICU. The mortality and morbidity of such infections are substantial. These processes also put economic strain on the healthcare system. Additionally, the continued spread of antimicrobial resistance has made it more challenging both to prevent and treat severe infection. Until recently, there were few well-done trials addressing infection among the critically ill. However, over the last year, six important randomized studies have dealt with a range of topics at the intersection of infectious diseases and critical care. Our goal is to review these reports in order to clarify their major findings, significance, strengths, weaknesses, and clinical applications. Specifically, we explore and discuss six trials conducted in the areas of (1) prevention, (2) the present use of standard antimicrobials, and (3) novel adjunctive and antibiotic treatments. Through highlighting these trials, we hope to help clinicians apply their important findings in an evidence-based fashion at the bedside. It is through the application of key evidence that both infectious disease practitioners and intensivists can improve patient outcomes.

## 1. Introduction

Severe infection leads to substantial morbidity and mortality. For example, community-acquired pneumonia (CAP) remains a leading cause of hospitalization with an estimated 7–10% of patients dying while hospitalized [1]. Likewise, in community-onset septic shock, the average length of stay (LOS) in the United States approaches 10 days and results in costs exceeding USD 75,000 per case [2]. Crude in-hospital mortality rates for septic shock, moreover, continue to exceed 35% [3]. The burden of nosocomial infections is more staggering. Ventilator-associated pneumonia (VAP) represents a major complication of respiratory failure requiring mechanical ventilation (MV). Although the attributable mortality of VAP is likely low, VAP prolongs the LOS in the intensive care unit (ICU) substantially and is estimated to lead to added costs of over USD 100,000 per event [4]. Highlighting the burden of VAP, a recent analysis suggests that VAP complicating Severe Acute Respiratory Syndrome Coronoavirus-2 (COVID-19)-associated respiratory failure led to as many deaths as did COVID-19 directly [5].

The surge in antimicrobial resistance (AMR) over the last decade has only complicated the care of those suffering an acute infection. AMR has become a challenge both in Gram-positive and Gram-negative bacteria. The problem is so prevalent that the term “multidrug resistant” (MDR) has become nearly meaningless. Rather, now, phrases such as “pan-resistant” (PDR), “extensively drug resistant” (XDR), and “difficult to treat” (DTR) are more useful in that they capture the dilemma facing the clinician: how does one ensure the patient receives initially appropriate antimicrobial therapy while being a responsible steward of antibiotics? Multiple analyses have demonstrated that initially appropriate in vitro therapy is a central determinant of outcomes in severe infection [6,7]. Nonetheless, in an era of substantial AMR, it becomes difficult to balance the competing pressures of creating treatment protocols that lead to high rates of initially appropriate antimicrobial treatment while respecting the principles of antibiotic stewardship—all the while focusing on not fostering further resistance.

Additionally, the costs of AMR are excessive. One prior report indicates that AMR costs European Union nations more than EUR 9 billion annually [8]. The Centers for Disease Control and Prevention similarly calculated that AMR results in nearly USD 20 billion in added expenses to the US healthcare system [9]. At the patient level, Kingston and co-workers in a meta-analysis of studies examining the burden of AMR suggest that among Staphylococcus aureus bloodstream infections (BSIs), the presence of methicillin resistance (MRSA) was associated with an extra LOS of approximately 2.5 days [10]. Resistance to third-generation cephalosporins in Escherichia coli infections lead to a similar increase in LOS [10]. Poudel et al., relying on a slightly different meta-analytic approach, calculated that AMR in severe infection increased LOS by more than 7 days while also nearly doubling the risk for hospital mortality. They also documented that those with infections caused by AMR pathogens were approximately 50% more likely to be readmitted to the hospital [11].

In short, there is an urgent need to improve outcomes for patients suffering from significant infection, especially those who either require ICU admission for the acute process or who develop infectious complications while hospitalized in the ICU. In the past, the lack of well-done trials at the intersection of critical care and infectious diseases has hampered efforts to reduce the mortality and morbidity related to severe infection. In the absence of evidence, physicians have little guidance on how to modify their practice. Fortunately, a number of recent clinical trials have addressed important topics both in acute infection and in the area of AMR. Specifically, in 2023, six key trials reported outcomes on a range of interventions varying from novel approaches to dosing standard antibiotics to the efficacy of novel treatments. Similarly, new reports address preventive efforts in the ICU, the potential for adjunctive therapies for those suffering from severe CAP, and the safety of certain traditional antibiotics. Our goal is to review these trials so as to not only describe the questions they focus on and their findings but also to place each in context so their important results can be easily assessed and potentially adopted by bedside practitioners.

## 2. Materials and Methods

We selected important randomized trials published in 2023 that deal with infection in the ICU. We chose articles based on our assessment of their significance and likely impact on clinical practice.

## 3. Prevention

### 3.1. VAP

As noted above, VAP represents a major nosocomial complication of MV. Defined as a new pneumonia arising after more than 24–48 h of MV, the incidence of VAP ranges between 2 and 30 cases/1000 MV days [12]. Rates of VAP vary significantly based on multiple variables such as the ICU type (e.g., medical vs. surgical), the patient’s underlying co-morbidities, and the preceding duration of MV. Despite heterogeneity in the prevalence of VAP in various ICUs, VAP remains a global challenge and is seen in both higher-income as well lower- and middle-income nations. AMR pathogens often cause VAP with *Pseudomonas aeruginosa* (PA), carbapenem-resistant Enterobacterales (CRE), carbapenem-resistant *Acinetobacter baumanii* (CRAB), and MRSA being the most worrisome organisms. Although these bacteria can lead to VAP at any point in the course of MV, they are most often isolated in late onset (>3 days of MV) VAP [12]. Early onset VAP tends to result from infection with far less DTR microorganisms [12].

Pathophysiologically, VAP arises as a direct complication of MV and involves two basic steps: upper airway colonization and microaspiration. The placement of an endotracheal tube (ETT) fosters colonization of the upper airway with potentially pathogenic bacteria. Issues with the ETT and its design allow sections filled with bacteria from the oropharynx and upper airway to be aspirated into the lower airway [13]. These then serve as a nidus for a new infection. In light of this, many preventive options exist for addressing VAP. Keeping the patient’s head of the bed elevated and using orogastric (as opposed to nasogastric) tubes for feeding help to prevent VAP by addressing both airway colonization and microaspiration. Routine toothbrushing, moreover, has a significant impact on the likelihood of VAP developing [14]. Likewise, reducing the duration of MV necessarily decreases the potential for VAP. Therefore, guidelines recommend routine daily awakening and spontaneous breathing trials.

Hence, Ehrman and co-workers, completed a randomized controlled trial (RCT) exploring the utility of inhaled amikacin at preventing VAP in patients requiring MV [15]. Specifically, in this well-done double-blind study, the authors randomized patients to receive either inhaled daily amikacin (20 mg/kg of ideal body weight) or placebo for 72 h. The incidence of VAP served as the primary endpoint and the investigators had to initiate nebulized amikacin within 96 h of the start of MV. Unlike earlier studies, they employed a rigorous definition of VAP that required quantitative respiratory cultures and an evaluation by a blinded adjudication panel. They also strived to maintain the blinding of investigators by going so far as to physically conceal the contents of each nebulization. (Table 1)

The final study cohort included 847 subjects (417 randomized to amikacin) with a median age of 62 years [15]. The majority of patients were treated in medical ICUs and were moderately ill based on the severity of illness scores. Those receiving amikacin were significantly less likely to suffer VAP. Specifically, 22% receiving placebo vs. 15% of persons treated with inhaled amikacin were diagnosed with VAP (Table 1). When adjusting for the duration of MV, amikacin reduced the rate ratio for VAP substantially: 0.68 (95% confidence interval [CI]: 0.49–0.94, *p* = 0.004) [15]. The positive impact of amikacin persisted, and was of the same overall magnitude, when the endpoint was limited to only Gram-negative VAP.

With respect to key secondary outcomes, prophylactic inhaled amikacin had no clear impact. Not surprisingly, since there is likely no attributable mortality related to VAP, the authors saw no difference in 28-day mortality between the two groups. Unfortunately, however, reliance on inhaled amikacin failed to affect the duration of MV, ICU LOS, or hospital LOS [15]. More surprisingly, there was no difference in either attempted spontaneous breathing trials or subsequent antibiotic utilization.

The study was likely underpowered to address whether this tactic for VAP prevention can alter the duration of MV. However, if VAP was noted very often in the placebo group (>20% of subjects), why did inhaled amikacin fail to increase the likelihood of a spontaneous breathing trial (indicating potential readiness to come off MV) or to decrease antimicrobial use? The likely answer is a function of two factors. First, the overall duration of MV in the study, including the time from the start of MV to randomization, was approximately 12 days [15]. Therefore, 3 days of inhaled amikacin may be insufficient to truly impact clinically meaningful variables. Readers should note that the study authors selected the 3-day duration based mainly on practical factors and a belief as to when the incidence of VAP would peak. Second, it is unclear what proportion of the VAP that was prevented was due to MDR or DTR pathogens. Put simply, although Ehramn et al. completed a well-done and rigorous trial, they essentially prevented a “distinction without a difference”. Alternatively, based on their findings, one would require a trial of over 10,000 patients to potentially show that inhaled prophylactic amikacin reduced the duration of MV by one day. Thus, one has to question, even in a best-case scenario, how important this intervention could ever be.

### 3.2. S. aureus

In contrast to preventive measures targeting specific syndromes such as VAP, researchers have also focused on addressing key pathogens such as *S. aureus*. *S. aureus* generally, and MRSA, specifically, represent important bacterial challenges for the critically ill [16]. These organisms cause a range of infections including VAP, BSI, and cellulitis. Presently, nearly one in four infections treated in the ICU arise because of *S. aureus* [17]. This high prevalence illustrates the substantial burden that *S. aureus* places on the healthcare system.

Although MRSA accounts for more than 60% of all *S. aureus* isolated in ICU patients, general rates of infection due to both MRSA and methicillin-susceptible *S. aureus* (MSSA) have declined over the last 15 years [16]. Multiple factors account for the decline in MRSA and MSSA infections. For example, specific preventive care bundles for both ventilated patients and for subjects with central lines have generally reduced, respectively, the prevalence of both VAP and CLABSIs—where *S. aureus* is a leading culprit pathogen. In addition, most ICUs across the globe employ routine chlorohexidine (CHG) bathing to facilitate skin decolonization so as to reduce *S. aureus* infections.

Approximately a decade ago, investigators completed the central trial supporting this paradigm. In the landmark REDUCE MRSA study, researchers demonstrated that a universal decolonization strategy relying on the combination of CHG and nasal mupirocin (as opposed to a targeted approach focusing on patients known to be colonized) reduced rates of both MRSA clinical cultures and BSIs [18]. Despite these findings, it is important to note that some controversy remains regarding the value of isolated regular CHG bathing. Noto and coworkers, in a single-center RCT of over 10,000 patients, reported that CHG had no impact on the incidence of either VAP, CLABSI, or catheter-associated urinary tract infection (cUTI) [19]. Additionally, CHG did not affect the frequency of BSI or Clostridium difficile associated diarrhea (CDAD). This study, however, did not include nasal decolonization, as was done in the REDUCE MRSA trial [18,19].

Hence, it would appear that mupirocin application is crucial to efforts at decolonization and infection control. Adoption of mupirocin, though, lags behind reliance on CHG bathing. A recent survey reveals that less than 40% of US hospitals routinely utilize mupirocin in the regular nursing care of ICU patients [20]. One specific concern related to broad utilization of mupirocin relates to the potential for emergence of resistance. Several prior analyses suggest that resistance can develop with regular application of this agent [21,22]. A meta-analysis by Dadashi et al. of various studies exploring ways to eradicate MRSA carriage, for instance, determined that resistance can evolve in between 1 and 9% of patients [22].

It was because of this concern that researchers undertook to determine if an antiseptic, iodophor, could replace mupirocin as a tool for infection prevention. Given its nature as an antiseptic, the emergence of resistance is much less of a worry. Iodophor also has general in vitro activity against *S. aureus* and has been evaluated as a preventive option in earlier studies [20]. Its efficacy in large, non-selected populations, though, has not been systematically determined. However, in a large multicenter RCT enrolling over 800,000 subjects across 137 hospitals, Huang and co-workers randomized ICU patients to 5 days of nasal topical mupirocin or iodophor (Table 1) [20]. At baseline, institutions were utilizing mupirocin. Therefore, these researchers examined rates of clinical MRSA and MSSA cultures during a baseline period of 24 months. This established each ICU’s current burden of *S. aureus*. ICUs were then randomized to either continue applying mupirocin or to switch to iodophor. In order to limit any cofounding effects from prior reliance on mupirocin, the investigators did not begin to collect data until after a 6-month run-in period. This was followed by 18 months of observation and evaluation. During the trial, investigators conducted intensive education for nurses and clinicians while also auditing compliance with the designated interventions. Again, these interventions were undertaken within a background of all patients undergoing CHG bathing.

**Table 1 biomedicines-12-00562-t001:** Summary Of Reviewed Clinical Trials.

Authors	Reference	Clinical Focus	Clinical Question	Trial Design	Sample Size	Conclusion
Ehrman et al.	[15]	Prevention	Does 3 days of nebulized amikacin prevent VAP?	DBRCT	850 MV patients	Amikacin reduces the risk for VAP: 15% with amikacin vs. 22% with placebo (*p* < 0.05). No impact on duration of MV, LOS
Huang et al.	[20]	Prevention	Is nasal iodophor non-inferior to mupirocin at preventing clinical cultures with *S. aureus* (in a background of CHG bathing)?	Pragmatic, cluster RCT	801,688 ICU admissions	Iodophor was inferior to mupirocin at preventing clinical cultures with *S. aureus*. It also had no impact on cultures revealing MRSA or ICU-attributable BSIs.
Monti et al.	[23]	Treatment	Does continuous vs. intermittent administration of meropenem reduce either mortality to subsequent cultures with PDR or XDR pathogens?	DBRCT	607 patients with severe sepsis	Continuous infusion had no impact on either mortality or the emergence of resistance
Qian et al.	[24]	Treatment	Does piperacillin-tazobactam increase the risk for AKI compared to cefepime?	Open label RCT	2511 patients requiring an anti-pseudomonal antibiotic	There was no difference in the rate of AKI between the two agents. There were more neurologic complications (delirium/coma) with cefepime.
Dequin et al.	[25]	Novel approach	Do adjunctive corticosteroids in severe CAP reduce mortality?	DBRCT	800 subjects with severe CAP (but not in septic shock)	Corticosteroids reduce the risk for death significantly (6.2% vs. 11.9%, *p* = 0.006). The NNT to save one life equaled 18.
Kaye et al.	[26]	Novel approach	Is a novel agent, sulbactam-durlobactam, non-inferior to colistin for mortality	DBRCT	128 persons with CRAB infections identified by a rapid diagnostic.	Sulbactam-durlobactam was non-inferior to colistin with respect to mortality in the treatment of severe CRAB infections. Clinical cure rates were significantly higher with sulbactam-durlobactam and this agent resulted in less AKI than colistin.

Abbreviations: AKI—acute kidney injury, BSI—bloodstream infection, CAP—community-acquired pneumonia, CHG—chlorhexidine, CRAB—carbapenem-resistant *A. baumanii*, DBRCT—double blind randomized trial, ICU—intensive care unit, LOS—length of stay, MRSA—methicillin-resistant *S. aureus*, MV—mechanical ventilation, NNT—number needed to treat, VAP—ventilator-associated pneumonia.

Most patients were cared for in mixed medical–surgical ICUs, and none of the participating hospitals were academic medical centers. The final analysis population included nearly 1.5 million patient days of either mupirocin or iodophor [20]. Although designed as a non-inferiority trial to demonstrate that one could safely replace mupirocin with iodophor, iodophor was strikingly inferior to mupirocin. For example, in hospitals that remained relying upon mupirocin, the rate of clinical MRSA and MSSA cultures remained similar between the baseline and intervention periods. However, with iodophor, there were more positive clinical *S. aureus* cultures after the switch away from mupirocin. As the authors write, “The relative hazard of *S. aureus* clinical cultures was significantly higher by 18.4%” with iodophor [20]. For BSI rates, there was no difference. Readers should note, though, that BSIs were a generally relatively rare occurrence. Various alternative analyses looking at subpopulations where adherence was higher further underscored that iodophor performed inferiorly to mupirocin. Impressively, over the duration of the entire trial, including baseline, run-in, and observation periods, the effect of mupirocin appeared durable (Table 1).

Although it is evident that hospitals cannot adopt iodophor for their infection prevention protocols, this trial re-confirms the value of combined CHG bathing and nasal mupirocin. The large sample size coupled with the long duration of observation should assure hospital leaders that they should stress this paradigm in their institutions—even more so now given that in the United States hospital-onset MRSA bacteremia is viewed as a key quality metric. Unfortunately, investigators did not collect data regarding the emergence of mupirocin resistance [20]. Hence, questions regarding the frequency of this outcome and its clinical significance remain unanswered. One should note that compliance in the mupirocin arm was higher than in ICUs randomized to iodophor. Some have suggested that this might explain the inferiority of iodophor. However, when researchers restricted their analysis to subjects where compliance with both agents was high, they continued to note the inferiority of iodophor [20]. This finding should reassure potential skeptics. In sum, CHG bathing coupled with nasal mupirocin represents a key standard of care for prevention in critically ill patients.

## 4. Current Antibiotics

### 4.1. Extended Infusions

One approach for overcoming AMR focuses on developing new antimicrobials specifically targeted against DTR pathogens and their specific mechanisms of resistance. This tactic is fraught with risk, and it often takes a decade to bring a new antimicrobial to the bedside. Alternatively, one can attempt to overcome in vitro resistance by leveraging well known principles of pharmacokinetics and pharmacodynamics (PK-PD). For beta-lactams, cephalosporins, and carbapenems, the relationship between the area under the dosing curve but above the minimum inhibitory concentration (AUC/MIC) for the pathogen of interest is the main PK-PD target that correlates with clinical outcomes [27]. Presently, most of these types of agents are infused rapidly (e.g., over 30 min), which leads to a high peak concentration in the blood (or target tissue) but limits the AUC/MIC. One simple way to better attain this AUC/MIC ratio target is to extend the infusion time. By extending the antimicrobial infusion over several hours, one necessarily maintains the antibiotic concentration above the MIC for a longer period. With this strategy, one might be able to utilize standard, less expensive agents to treat pathogens with higher MICs. Reflecting the importance of this approach, many novel antibiotics (e.g., Ceftolozone-tazobactam, Ceftazadime-avibactam) are designed to be given as extended infusions [28,29].

Previously, several randomized trials have tested the hypothesis that extended or continuous infusion of certain agents improves outcomes. These studies have produced conflicting results. For example, Abdul-aziz et al. showed, in an RCT including 140 patients with severe sepsis, that continuous infusion led to fewer deaths than did standard, intermittent infusion (34% vs. 56%, *p* = 0.01) [30]. Dulhunty and co-workers, though, in a larger trial (n = 432) noted that continuous infusion had no impact on any measure, be it mortality, LOS, or organ failure [31]. A meta-analysis of multiple trials exploring this question concluded that continuous or extended infusion likely benefits patients, but the magnitude of the impact was potentially small and moderated by a number of factors including severity of illness and the type of infecting pathogen [32]. A separate meta-analysis utilizing different methodologic approaches reached a similar conclusion [33].

In light of this controversy, the investigators of the MERCY RCT sought to better assess if and for whom continuous infusion might offer a benefit (see Table 1). MERCY enrolled patients with sepsis or septic shock across 31 ICUs in Europe and Asia [23]. All enrolled received a 1 g loading dose of meropenem. Then, in a blinded fashion, patients were randomized to either standard infusion meropenem (1 g every 8 h over 30–60 min) or continuous meropenem (3 g given over 24 h). To guarantee that blinding was maintained, all subjects were also given dummy agents as well. The primary endpoint consisted of a combination of either mortality at day 28 or the subsequent isolation of an PDR or XDR pathogen [23].

The final study population included over 600 patients, making MERCY the largest trial to date of continuous infusion antibiotics. More than three quarters of patients required MV and more than 60% met criteria for septic shock. Pneumonia represented the most prevalent underlying infection. Moreover, approximately 10% of patients had concomitant COVID-19 infection. The most commonly recovered bacteria included Klebsiella species, *P. aeruginosa*, and *E. coli*. One-third of isolates displayed in vitro resistance to carbapenems [23].

The primary endpoint occurred in 47% of subjects randomized to continuous infusion and in 49% of those given standard dosing meropenem (Table 1). Unfortunately, continuous infusion, in other words, offered no clinical benefit [23]. Furthermore, neither subcomponent of the primary endpoint differed based on the approach to meropenem infusion. Additionally, no secondary endpoint, such as ICU LOS, 90-day survival, or duration of antibiotic utilization, varied based on form of meropenem administration.

Does MERCY conclusively close the door on the utility of continuous antibiotic infusion? Despite MERCY’s strengths, it was likely underpowered to provide definitive proof on this topic. Recall that although the trial enrolled over 600 patients, only approximately 200 had a documented bacterial infection—and only a third of whom had a carbapenem-resistant isolate [23]. If there were a population where continuous infusion were to prove helpful it would be in this very subgroup of subjects as these are the patient’s where extended infusion might allow one to overcome the higher MICs associated with resistance because the drug concentration would be kept above the MIC for a longer period of time. This would be particularly true in septic patients where there would likely be variable subpopulations of bacteria with differing MICs. Biologically, and in light of PK-PD principles, if an infection arose due to a pan-sensitive *E. coli*, for example, there is little reason to presume that continuous infusion would offer a benefit. To this point, in the supplementary data accompanying the paper, the authors present an analysis of the 110 subjects who had an MDR bacteria isolated during the 48 h either before or after enrollment [23]. In this subgroup, the primary endpoint occurred in 46% of the continuous infusion arm vs. 64% of the traditional approach. This 18% difference approached statistical significance (*p* = 0.06) [23]. Of course, all post hoc subgroup analyses must be viewed as only hypothesis generating. Nonetheless, this observation suggests that the challenge may be in identifying the correct population that might benefit from continuous infusion. In addition, MERCY only examined meropenem. It is unclear if the findings would have been similar if the authors had studied different antibiotics such as cefepime or piperacillin-tazobactam. Similarly, continuous infusion essentially represents a surrogate effort to optimize PK-PD. Without direct therapeutic drug level monitoring (TDM), though, one cannot be certain that the specific PK-PD target was actually reached.

Although the findings from MERCY indicate that clinicians need not routinely employ continuous infusion at present, more conclusive information will come from the BLING III trial [34]. BLING III is a multinational unblinded RCT of continuous vs. intermittent infusion of either meropenem or piperacillin-tazobactam. Like MERCY, BLING III will enroll patients with severe sepsis and septic shock. However, the planned sample size is far greater. Researchers aim to include 7000 patients across over 70 ICUs in BLING III [34]. Thus, by design, BLING III will address several of the important limitations of MERCY. Furthermore, BLING III will include a large subpopulation that will undergo TDM to specifically examine PK-PD target attainment and outcomes. The results from BLING III are expected in 2024.

### 4.2. Nephrotoxicity

As alluded to earlier, patients with suspected infection require the prompt initiation of appropriate antimicrobial treatment. In many hospitals, this generally leads to the near reflexive administration of either piperacillin-tazobactam or cefepime. Despite their widespread use, safety concerns exist with both molecules. The potential increased risk for acute kidney injury (AKI) associated with piperacillin-tazobactam, especially when given concomitantly with vancomycin, has resulted in prescribers utilizing more cefepime [35]. However, cefepime’s presumed lower risk for nephrotoxicity may be rivaled by its potential for neurotoxicity [36]. A 2018 meta-analysis of 24,799 patients examined AKI related to the combination of vancomycin and piperacillin-tazobactam. The authors concluded there was an increased risk of AKI with the combination [37]. Most of the concern regarding the risk for AKI with piperacillin-tazobactam generally derives from retrospective studies that have multiple limitations such as selection bias and confounding by indication. In the absence of comparative, prospective safety data, providers lack crucial information for making decisions on appropriate antibiotic choices. To this extent, high-quality data are imperative in guiding therapy and achieving optimal patient outcomes.

Recognizing this gap in knowledge, Quain et al. examined whether either piperacillin-tazobactam or cefepime increased the risk of AKI or neurological dysfunction in patients with suspected infection in the Antibiotic Choice on Renal Outcomes (ACORN) trial (Table 1) [24]. This single-center RCT studied 2511 patients with presumed sepsis. They were enrolled while in the emergency department (ED) or the ICU and had initial orders for piperacillin-tazobactam or cefepime within 12 h of hospital presentation. The highest stage of AKI or death by day 14 served as the primary endpoint. The number of days alive, days free of delirium or coma, and percentage of patients who experienced a major kidney event by day 14 were identified as secondary outcomes. In addition to receiving piperacillin-tazobactam or cefepime, approximately 80% of patients were treated with at least one dose of vancomycin. More than half of subjects were exposed to other potentially nephrotoxic agents. In the end, there was no difference in occurrence of the primary endpoint as a function of antimicrobial selection. The rates of AKI or death by day 14 did not significantly differ between arms. This finding was consistent in the subgroup of patients who concomitantly received vancomycin. Of note, those who received cefepime had higher rates of delirium or coma (20.8% vs. 17.3%) [24]. These findings demonstrate similar nephrotoxic risk between these two antibiotics but a higher risk of neurotoxicity in the cefepime group. While promising, one must be cautious when interpreting these data and extrapolating these findings to the ICU population.

In this study, only 329 critically ill subjects were randomized to either piperacillin-tazobactam or cefepime. Of all the subjects enrolled who ultimately had positive cultures, the majority did not develop AKI. In the cefepime group, 66.3% had no AKI compared to 66.2% for piperacillin-tazobactam (Table 1) [24]. This fact suggests that the use of piperacillin-tazobactam alone, or as combination therapy, is not independently associated with an increased risk of AKI, even in the cohort most at risk for AKI given their underlying disease severity.

Conversely, sepsis is responsible for 45–70% of AKI in critically ill patients and requires an antimicrobial course that well exceeds the 2–3-day median duration of treatment in ACORN [38]. The short duration of antimicrobial treatment raises the question of whether or not those enrolled truly had active infection. Furthermore, vancomycin-associated AKI is generally seen after 4–17 days of administration [6]. With a median duration of vancomycin administration of only 2 days, there may have been insufficient exposure for nephrotoxicity to potentially develop. The influence of other underlying co-morbidities and advanced therapies unique to this population must also be considered in weighing the absolute risk of AKI. Finally, the study was clearly underpowered to explore AKI in the critically ill.

The increased risk for neurotoxicity with cefepime is mildly concerning. Up to 80% of critically ill patients experience delirium during hospitalization [39]. Adding an agent that could compound this should give pause. While cefepime resulted in statistically significantly higher rates of delirium or coma, its use did not impact either mortality or LOS—leaving one to question its clinical significance. Furthermore, the subjectivity of the care team and the unblinding of patients may have created opportunity for bias, thereby possibly influencing the assessment of this safety endpoint.

The safety debate surrounding both AKI and neurotoxicity related to piperacillin-tazobactam and cefepime has proceeded for several years. To date, this is the first head-to-head RCT. The authors deserve praise for conducting a well-executed study, setting a precedent for future trials. By leveraging their electronic medical record to identify and randomize subjects, the investigators collected a large sample with data collected in real time. They also demonstrated that regardless of acuity, the studied drugs did not impact renal function. At its core, the optimal choice of empiric therapy must reflect a careful risk vs. benefit analysis, and this trial, despite some limitations, provides reassurance to providers.

## 5. Novel Paradigms

### 5.1. CAP

Corticosteroids (CS) have become a mainstay of therapy in refractory septic shock and in COVID-19 respiratory failure [40,41]. Directly addressing the significant inflammation that evolves in these syndromes improves patient-centered outcomes. Their role as adjunctive therapy in severe CAP, though, has remained more unclear. Some small trials have suggested that adding CS to antibiotics in CAP leads to faster symptom resolution and shorter hospital LOS [42,43]. These studies, however, did not routinely mandate the use of macrolides as part of the treatment regimen—raising concerns that CS might add little if a patient were already benefiting from the anti-inflammatory effects of a macrolide. Additionally, most of the earlier studies suggesting a benefit of CS in severe CAP have either suffered from limitations in trial design (e.g., open label as opposed to blinded) or been underpowered to address the key issue: does the use of CS in CAP reduce mortality? One recent large, double-blind RCT, however, revealed that, among nearly 600 subjects with CAP, 40 mg daily of methylprednisolone had no impact on survival [44]. This trial, also had many limitations and may have been underpowered [44].

It is in this background that Dequin et al. conducted the CAPECOD trial (Table 1) [25]. In contrast to many earlier studies, CAPECOD was a double-blind RCT enrolling patients in over 30 centers in France [25]. The authors enrolled patients with a clinical diagnosis of CAP that necessitated ICU care. Importantly, they excluded those (1) with influenza (as CS have been shown to worsen outcomes in influenza) and (2) with septic shock. Patients had to receive CS within 24 h of meeting criteria for severe CAP. Investigators administered 200 mg of hydrocortisone daily as a continuous infusion to those randomized to treatment with CS [25]. Hydrocortisone was given for a minimum of 4 days but could be extended based on a subject’s clinical status. CS were discontinued prior to ICU discharge.

Mortality at day 28 represented the primary endpoint while need for subsequent MV or treatment with vasopressors served as key secondary endpoints. The final cohort included 795 subjects [25]. Although all patients were in the ICU, 22% required invasive MV. Additionally, reflecting the exclusion of those in septic shock, only 11.5% of those enrolled were receiving vasopressors at baseline. To place these observations in context, persons treated in CAPECOD were clearly less severely ill than individuals participating in the VAP prophylaxis trial conducted by Ehlrman et al. discussed earlier [25]. With respect to the degree of inflammation among CAPECOD participants, nearly 70% had C-reactive protein levels exceeding 15 mg/dL [25].

Despite the trial being stopped early because of the pandemic, hydrocortisone treatment significantly led to fewer deaths. Approximately 12% of the placebo cohort died by day 28 compared to nearly 6% of those given CS (*p* = 0.006) [25]. This difference translates into a number needed to treat (NNT) to save one life of eighteen (Table 1). Very few interventions have ever been shown to have that magnitude of effect in the critically ill. This benefit persisted for up to 90 days. Both rates of subsequent MV via an ETT along with a new need for vasopressors were lower in persons randomized to therapy with hydrocortisone. Importantly, CS were well tolerated. They promoted neither nosocomial infections nor gastrointestinal bleeding. Given their nature, not surprisingly, persons in the CS arm required more insulin for hyperglycemia.

Unlike earlier studies in this area, most patients in CAPECOD received both initially appropriate therapy and concurrent treatment with a macrolide. These strengths underscore the importance of the results from CAPECOD. Furthermore, a culprit pathogen was isolated in 55% of trial participants [25]. Although potentially seen as a limitation, this level of organism recovery is consistent with the vast majority of studies into the epidemiology of CAP.

Why do the findings of CAPECOD differ so dramatically from the only other large trial exploring this issue? First, the prior study was conducted in US Veterans Affairs (VA) hospitals [25,44]. As a consequence, nearly all participants were male—while 30% of those in CAPECOD were female. Gender-based differences in critical care have been previously noted, and sex hormones may interact during severe infection to affect inflammation in such a way that CS may now be protective. This, though, seems unlikely as the results in CAPECOD did not appear to vary based on gender. Second, patients in CAPECOD received CS early in the course of their critical illness. For example, the median time to receiving CS after meeting disease severity criteria was only 15 h [25]. In the VA report, the window for CS administration was much longer and extended out to 96 h after presentation [44]. As with many treatments in the ICU, earlier intervention provides a more substantial benefit.

Another important distinction between the VA trial and CAPECOD involves the role for rescue CS in the event that septic shock developed [25]. In the VA study, where there was no benefit to CS in severe CAP, clinicians could prescribe rescue CS to persons in whom CAP progressed to septic shock. This option was not permitted in CAPECOD. If a subject was randomized to placebo and subsequently developed septic shock, they could not be given CS. In a post hoc analysis, Dequin et al. note that 9.7% of patients randomized to CS (n = 35) progressed to septic shock requiring vasopressors as opposed to 17.7% (n = 61) of those in the placebo arm [25]. One must wonder: if rescue CS were allowed, would this have attenuated the apparent mortality benefit of CS?

Furthermore, readers should note that the vast majority of the benefit of CS was restricted to persons with clear inflammation. In those with low C-reactive protein levels, the point estimate for the difference in mortality with CS was only 2.4% (as compared to 7.3% in the elevated C-reactive protein population) [25]. This does not mean that CS should be withheld if measures of C-reactive protein are normal. That would be a misinterpretation of the meaning of a small subgroup. However, it does underscore the biologic plausibility of the significant role CS can play as an adjunctive therapy in severe CAP.

Despite several of the concerns and limitations noted above, the preponderance of the data now suggests that clinicians must consider adjunctive treatment with CS in persons with severe CAP. More importantly, the decision on whether to administer CS must be made quickly.

### 5.2. Sulbactam-Durlobactam

Acinetobacter baumanii represents a noteworthy pathogen across the globe. In parts of Asia, South America, and Eastern Europe, *A. baumanii* causes not only hospital-acquired infections but also community-onset processes [45]. In the US, *A. baumanii* is less common and is generally recovered mainly in nosocomial infections such as VAP [46]. More significantly, the majority of *A. baumanii* display in vitro resistance to carbapenems (CRAB). As a result, few patients with infections due to CRAB receive initially appropriate antibiotic therapy. In one analysis, clinicians failed to prescribe timely, appropriate antibiotic treatment to approximately 80% of subjects with either CRAB pneumonia or sepsis [46]. Consequently, the crude mortality rate in CRAB infections is high. Moreover, delays in appropriate therapy for CRAB result in excess costs of approximately USD 2000 per day of delay (2023 USD) [46].

At present, few treatment options exist for CRAB. Polymixin-based regimens remain the backbone of therapy in many institutions. Unfortunately, polymyxin b and colistin have major limitations. These agents have very poor lung penetration; thus, their utility in pneumonia is not well established. In addition, the optimal dosing paradigm remains unclear. Although various simulations and pharmacokinetic analyses suggest potential optimized regimens, no RCT-based evidence truly supports the currently recommended doses for these agents [47]. Finally, these antibiotics are major nephrotoxins. Multiple studies reveal that the use of polymyxin b and/or colistin independently increases the risk for renal failure and the need for acute renal replacement therapy.

Most new antimicrobials lack activity against CRAB. One, cefiderocol, because of its unique mechanism of action, is in vitro active against CRAB. There were few infections due to CRAB, however, in the multiple RCTs that led to the regulatory approval of this agent [48,49]. Additionally, in an open-label trial of cefiderocol—where CRAB caused many of the infections—and where colistin was the most often used comparator, investigators noted increased mortality in patients treated with cefiderocol [50]. Hence, there is an urgent need for a safe and effective agent for treating severe CRAB infections.

Sulbactam-durlobactam is a combination of a beta-lactam and a novel beta-lactamase inhibitor. Used together, these agents possess potent in vitro activity against carbapenem-susceptible *A. baumanii*, generally, and CRAB, specifically. In ATTACK, researchers aimed to demonstrate the clinical utility of sulbactam-durlobactam (Table 1) [26]. In contrast to several other studies described above, ATTACK was designed as a non-inferiority, rather than superiority, trial [26]. In other words, investigators in this industry-sponsored project aimed to show that sulbactam-durlobactam resulted in similar clinical outcomes vs. the comparator, colistin. More specifically, patients were randomized in a blinded fashion to receive either sulbactam-durlobactam (1 gm of each) infused over three hours four times a day or colistin (2.5 mg/kg) every 12 h [26]. Subjects had to be treated for a minimum of 7 days, and treatment could not be extended beyond 14 days. All patients were treated with background imipenem-cilistatin. Unlike many earlier studies, and in an effort to ensure patients were more likely to receive initially appropriate therapy, the protocol required use of a rapid diagnostic to identify patients with *A. baumanii* infections. Furthermore, rather than exclude individuals who either could not tolerate colistin or whose pathogen was resistant to colistin, these persons were continued on sulbactam-durlobactam in a non-randomized fashion (referred to as part B of the trial) [26]. Mortality at day 28 served as the primary endpoint.

The final analysis population in the main cohort included 128 patients with CRAB [26]. Although a small trial relative to others reviewed above, this represents one of the larger studies addressing CRAB. Roughly 95% of patients suffered from some form of pneumonia and VAP accounted for nearly half of all cases [26]. The generally low severity of illness scores among the subjects indicates that the population was not severely ill. Of note, one third of the infections were polymicrobial. For the primary endpoint, the 28-day mortality rate with sulbactam-durlobactam equaled 19.0% as compared to 32.3% for colistin. Hence, the study met its primary endpoint for non-inferiority (Table 1) [26]. Readers should note that the 95% CI around the 13.3% difference in mortality barely crossed 0%; however, given the study’s non-inferiority design, even if the difference did not cross 0%, one could not logically conclude superiority [26]. Strikingly, clinical cure rates were significantly higher in persons randomized to sulbactam-durlobactam (62% vs. 40%, *p* = 0.02), as was mortality at day 14 [26].

Unsurprisingly, sulbactam-durlobactam resulted in far less nephrotoxicity than did colistin. Only 13% of patients in the sulbactam-durlobactam arm met RIFLE criteria for acute kidney injury compared to 38% of patients in the colistin arm (Table 1). In general, sulbactam-durlobactam was otherwise well tolerated. Among those in the non-randomized part B of ATTACK (n = 28), 16 subjects had isolates resistant to colistin. The crude mortality rate at day 28 among part B participants equaled 18%, and the clinical cure rate was 71% [26].

Despite demonstrating the efficacy of sulbactam-durlobactam, ATTACK raises a number of issues and has several important limitations. First, most patients were enrolled in sites outside of North America and Western Europe. This issue, along with the fact that few subjects were in shock or had bacteremia, likely limits the generalizability of the study’s findings. Relatedly, few patients were immunosuppressed. These concerns are made particularly evident given the low mortality rate in both arms of the trial. Historically, clinical trials with colistin for A. baumanni have noted mortality rates of approximately 50% [51,52]. Second, one should realize that although there was less nephrotoxicity with sulbactam-durlobactam, most of the difference is confined to more mild forms of renal injury. There was no difference in the need for acute renal replacement therapy—which one might have predicted, again, based on earlier analyses of colistin utilization. Third, confusion exists regarding the need for concomitant use of imipenem-cilistatin with sulbactam-durlobactam. Combination therapy was given to all subjects in ATTACK so as to guarantee that individuals with polymicrobial infections would receive initially appropriate therapy (as sulbactam-durlobactam is designed to specifically address only A. baumanni) and was continued in order to prevent accidental unblinding (should one investigator opt to stop it while others did not). Certainly, the combination of either agent in the study with a carbapenem may result in in vitro synergy. However, this in vitro activity has been shown specifically in A. baumanni not to affect clinical outcomes [53]. Hence, as regulatory authorities concluded, one need not combine sulbactam-durlobactam with a carbapenem in clinical practice. In sum, sulbactam-durlobactam represents a novel agent for treating CRAB. Future trials will be needed to elucidate its specific role and value.

## 6. Conclusions

AMR will remain a major threat to critically ill patients for years to come. Whether seen in the setting of pneumonia, BSI, or septic shock, MDR and DTR pathogens are likely to become more prevalent and thus present more of a danger to those least able to respond to such severe insults. Fortunately, over the last year, several key trials have addressed crucial questions for the prevention and treatment of severe infections in the ICU. Although some of these studies were positive while others were negative, each focused on an important question and was conducted in a rigorous manner. All also raise additional hypotheses that require further evaluation. Irrespective, it is only through embracing this emerging evidence and struggling to apply it at the bedside that clinicians will be able to improve outcomes for critically ill infected patients.

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
