# Peer review of "At the Intersection of Critical Care and Infectious Diseases: The Year in Review"

_biomedicines, 2024, doi:10.3390/biomedicines12030562_

Round 1
Reviewer 1 Report
Comments and Suggestions for Authors
This is a nice and pertinent review. The authors discuss the reviewed studies correctly, and the reader is able to grab the significant findings put in context.
Some questions for the authors to address for the sake of clarity:
1. “The positive impact of amikacin persisted when the endpoint was limited to only Gram-negative VAP” (line 127-128)
Did the effect only persist, or did it improve when analyzing only Gram negatives?
2. Could you comment on whether the authors of the CT do provide any explanation for their choice of such a short intervention (line 141), considering the usual duration of mechanical ventilation associated with VAP development?
3. “antiseptic, emergence of resistance is not a worry” (line 188)
This statement is not accurate. Take, for instance, van Dijk HFG, Verbrugh HA; Ad hoc advisory committee on disinfectants of the Health Council of the Netherlands. Resisting disinfectants. Commun Med (Lond). 2022 Jan 11;2:6. doi: 10.1038/s43856-021-00070-8. PMID: 35603291; PMCID: PMC9053202.
4. Do the authors of the RCT provide a rationale for using iodophor (line 187)? It would be interesting for the readers to get some data on previously known activity of iodophor against S aureus to put this trial in context.
5. “In sum, CHG bathing coupled with nasal mupirocin represents a key standard of care for prevention in critically ill patients” (line 266). Were CHG bathing used together with mupirocin or iodophor in the analyzed trial? If not, this sentence should be modified to adjust to the results of this particular RCT.
6. “If there were a population where continuous infusion were to prove helpful it would be in this very subgroup of subjects” (line 285) This statement is counterintuitive. Could you further elaborate in the reason why this particular population would be the one to benefit from extended infusion?
7. Extended infusion is also meant to overcome the altered antibiotic distribution encountered in septic patients. Could you please comment on this?
8. It would be interesting to further question pip-tazo + vanco nephrotoxicity citing this recent research: Miano TA, Hennessy S, Yang W et al. Association of vancomycin plus piperacillin-tazobactam with early changes in creatinine versus cystatin C in critically ill adults: a prospective cohort study. Intensive Care Med 2022; 48: 1144–55. https://doi.org/10.1007/s00134-022-06811-0
9. “those who received cefepime had higher rates of delirium or coma (20.8% vs 17.3%)” (line 343 ) Was this difference significant?
10. Line 20: “ the present use standard antimicrobials”. Do you mean “the present use OF standard antimicrobials”?
11. Line 69: the lack well done trials. Do you mean “the lack OF well done trials”?

Author Response
We thank the reviewer for the comments and have addressed them - in so doing we believe we have improved the paper. The responses are detailed below.
- We appreciate the reviewer’s point and have amended the sentence.
- We agree with the reviewer and have added language clarifying this point.
- We note the reviewer’s observation and have added modifying language.
- The reviewer makes an excellent point and we have expanded the comments regarding iodophor.
- Yes in this trial ALL patients undergo CHG bathing – we have added language to make this clear.
- We have clarified this point regarding extended infusion.
- We believe our point about MICs made in response to point 6 address this concern.
- We thank the reviewer for pointing out this article, but we respectfully disagree about the need to cite it since the ACORN trial is a definitive RCT while the study suggested is only an observational trial.
- Yes, the difference was statistically significant and we have added clarifying language.
- We thank the reviewer for noting our typo and have corrected it.
- We thank the reviewer for noting our typo and have corrected it.
Reviewer 2 Report
Comments and Suggestions for Authors
Six important randomized studies have dealt with a range of topics at the intersection of infectious diseases and critical care. Authors reviewed these reports in order to clarify their major findings, significance, strengths, weakness, and clinical applications. Specifically, they explored and discussed six trials conducted in the areas of 1) prevention, 2) the present use standard antimicrobials, and 3) novel adjunctive and antibiotic treatments. Through highlighting these trials, they hope to help clinicians apply their important findings in an evidence-based fashion at the bedside.
Although this review is well written and potentially interesting, several issues arise.
Abstract needs take home message.
Figure and table which suggest and explain author’s conclusion may be helpful.
Title may be helpful to include AMR.
3. Prevention: Is there only VAP?
3.2. S. aureus: Is there only S. aureus?
5.2 CRAB: I s there only CRAB?
Author Response
We appreciate the opportunity to respond to the reviewer's comments. In so doing we hope we have clarified matters and improved the manuscript. Our point by point responses are below.
- We appreciate the reviewer’s kind words regarding our paper.
- We have added a take home message to the abstract
- We respectfully disagree with the reviewer as for the need for another table as we believe the current table presents the key findings of the trials – adding language that reflects our editorial opinions would be somewhat duplicative.
- Again, we respectfully disagree regarding the title and the need to add AMR – in fact three of the studies we review (eg, CAPECOD, ACORN, the use of Amikacin for VAP prevention) do not specifically focus on AMR infections.
- As we describe in the introduction, ee picked well done RCTs done in the last year to review– only one addressed VAP prevention – and the Amikacin inhaled trial only focused on VAP – it did not address other pathogens.
- Again, see our comment above. Yes, the mupirocin vs iodophor trial only focused on S. aureus.
- Of large RCTs done in the last year for novel antimicrobials, only one was done and it dealt with only CRAB. The recent trial regarding cefepime and taniborbactam which was completed earlier but was not published until 2 weeks ago – hence making it outside the bounds of our review.